# Evaluation of Standard Precautions Compliance Instruments: A Systematic Review Using COSMIN Methodology

**DOI:** 10.3390/healthcare11101408

**Published:** 2023-05-12

**Authors:** Marzia Lommi, Anna De Benedictis, Barbara Porcelli, Barbara Raffaele, Roberto Latina, Graziella Montini, Maria Ymelda Tolentino Diaz, Luca Guarente, Maddalena De Maria, Simona Ricci, Dominique Giovanniello, Gennaro Rocco, Alessandro Stievano, Laura Sabatino, Ippolito Notarnicola, Raffaella Gualandi, Daniela Tartaglini, Dhurata Ivziku

**Affiliations:** 1UOC Care to the Person, Local Health Authority Roma 2, 00159 Rome, Italy; marzia.lommi@gmail.com (M.L.); barbara.porcelli@aslroma2.it (B.P.); barbara.raffaele@aslroma2.it (B.R.); graziella.montini@aslroma2.it (G.M.); ymelda.tolentinodiaz@aslroma2.it (M.Y.T.D.); siricci2017@gmail.com (S.R.); 2Fondazione Policlinico Universitario Campus Bio-Medico, 00128 Rome, Italy; a.debenedictis@policlinicocampus.it (A.D.B.); r.gualandi@policlinicocampus.it (R.G.); d.tartaglini@policlinicocampus.it (D.T.); 3Department of Health Promotion, Mother and Child Care, Internal Medicine and Medical Specialities, University of Palermo, 90128 Palermo, Italy; roberto.latina@unipa.it; 4Department of Biomedicine and Prevetion, University Tor Vergata, 00133 Rome, Italy; luca.guarente@gmail.com (L.G.); maddalena.demaria@outlook.it (M.D.M.); 5Department of Traslational Medical Sciences, University of Campania “Luigi Vanvitelli”, 81100 Caserta, Italy; dominique.giovanniello@unicampania.it; 6Centre of Excellence for Nursing Scholarship, Order of Nurses of Rome, 00136 Rome, Italy; genna.rocco@gmail.com (G.R.); alessandro.stievano@gmail.com (A.S.); laura.sabatino8@gmail.com (L.S.); ippo66@live.com (I.N.); 7Department of Nursing, Catholic University “Our Lady of Good Counsel”, 1000 Tirana, Albania; 8Department of Clinical and Experimental Medicine, University of Messina, 98100 Messina, Italy

**Keywords:** standard precaution, universal precautions, COSMIN, psychometric propriety, systematic review, scale, instrument, tool

## Abstract

Background: Standard precautions (SPs) are first-line strategies with a dual goal: to protect health care workers from occupational contamination while providing care to infected patients and to prevent/reduce health care-associated infections (HAIs). This study aimed at (1) identifying the instruments currently available for measuring healthcare professionals’ compliance with standard precautions; (2) evaluating their measurement properties; and (3) providing sound evidence for instrument selection for use by researchers, teachers, staff trainers, and clinical tutors. Methods: We carried out a systematic review to examine the psychometric properties of standard precautions self-assessment instruments in conformity with the COSMIN guidelines. The search was conducted on the databases PubMed, CINAHL, and APA PsycInfo. Results: Thirteen instruments were identified. These were classified into four categories of tools assessing: compliance with universal precautions, adherence to standard precautions, compliance with hand hygiene, and adherence to transmission-based guidelines and precautions. The psychometric properties of instruments and methodological approaches of the included studies were often not satisfactory. Only four instruments were classified as high-quality measurements. Conclusions: The available instruments that measure healthcare professionals’ compliance with standard precautions are of low-moderate quality. It is necessary that future research completes the validation processes undertaken for long-established and newly developed instruments, using higher-quality methods and estimating all psychometric properties.

## 1. Introduction

Standard precautions (SPs) are first-line strategies with a dual goal: to protect health care workers from occupational contamination while providing care to infected patients and to prevent/reduce health care-associated infections (HAIs) [1]. Adherence to standard precautions has been shown to reduce occupational risks for staff, morbidity and mortality in patients due to cross-transmission of infectious diseases, and healthcare costs [2,3].

SPs include the use of personal protective equipment (PPE), such as gloves, masks, safety glasses, hoods, closed shoes, and aprons, in situations involving potential contact with patients’ bodily fluids [4,5]. In addition, they include procedures such as hand washing, handling sharp materials, environmental recommendations, and handling materials that are in contact with the patient [4,5]. They differ from transmission-based precautions (TBPs), which are the second level of defense and are used when standard precautions are insufficient to prevent transmission of a known, suspected, or highly transmissible infection through the airborne route, droplets, or through direct or indirect contact with intact skin or contaminated surfaces [6,7].

The Centers for Disease Control and Prevention in 1985 [8], with the advent of HIV transmission, was the first to identify universal precautions (UPs), whose main purpose was to protect health care workers from exposure to potentially infectious biological substances (PIBs). As a result, the first tools for assessing compliance with UPs were born (such as Compliance with Universal Precautions). However, UPs were insufficient as biosecurity measures in patient-patient and health care provider-patient transmission, and so, in 1996, SPs were developed based on the principle that all body fluids, blood, secretions, excretions (except sweat), non-intact skin, and mucous membranes may contain transmissible infectious agents [1,9]. Although precautions have been widely instituted, recent studies show poor adherence to SPs by health care workers. Poor adherence is related to several factors, such as, for example, overconfidence, personal beliefs, emergency situations in care, excessive workload, unsatisfactory or outdated knowledge, experience in using PPE, and attitudes [1,10,11]. The latter, for example, affects compliance with standard or extensive precautions. Before deciding whether or not to follow precautions, healthcare providers rely on their perceptions of the severity of illness, the risk of infection, and the benefits or barriers of following precautions [12]. When then, in practice, health care workers are required to modify their behavior to reduce exposure to risk by changing their work habits, this is perceived as controlling, encounter resistance, and there may be a psychological tendency to cling to original knowledge and beliefs, which change very slowly and are persistent in the face of contrary evidence [13,14].

Over the years, several tools have been developed to assess compliance with standard precautions by health care providers. Two reviews have been published in the literature that examined tools to assess knowledge, attitudes, and compliance with standard precautions [15] and compliance with infection control practices [16]. The first literature review only covered instruments published up to 2011 to assess compliance with standard precautions in nurses [15]. The integrative review by Valim [16] included instruments published up to 2012 with the aim to report on the dimensions and contents of the instruments and on describing factors that influence compliance. These reviews included only instruments available up to 2012, paying attention to the behavior of general health care providers or specifically nurses. None of them followed the rigorous process of a systematic review or provided useful recommendations regarding the use of instruments in clinical practice. Since infection control in health care is an important quality indicator having critical effects on patients and professionals, identifying proper instruments to evaluate health care providers’ compliance with SPs becomes a priority topic for clinical practice and research. Additionally, the infection control process can be affected by different environmental and personal aspects asking for continuous guideline updates, suggesting this way for a periodical review of instruments and development of new ones. For all these reasons, we consider it useful and necessary to update a review on the available instruments, including the ones presented in the previous reviews and those developed afterward. Furthermore, it is considered essential to evaluate the development and validation studies of the instruments according to an approved and established methodology, such as the Consensus-based methodology Standards for the selection of health Measurement INstruments (COSMIN) [17,18], which evaluates the psychometric properties and quality of the studies according to scientific criteria.

Therefore, the aims of this review were: (1) to identify the instruments currently available for measuring healthcare professionals’ compliance with standard precautions; (2) to evaluate their measurement properties; and (3) to provide sound evidence for researchers, teachers, staff trainers, and clinical tutors to use when selecting instruments.

## 2. Materials and Methods

We carried out a systematic review to examine the psychometric properties of standard precautions self-assessment instruments in conformity with the COSMIN guidelines. The systematic review followed the Preferred Reporting Items for Systematic Reviews and Meta-Analysis (PRISMA) guidelines, and the review protocol was registered to PROSPERO with ID record number CRD42023408024. 

### 2.1. Search Strategy

The search was conducted on the databases PubMed, CINAHL, and APA PsycInfo until 1 March 2023. The PICOs formulated for the review were as follows: P-health care professionals’ compliance with standard precautions; I-identification and evaluation of development studies and validation of instruments that examine the compliance to the standard precaution and evaluation of the psychometric properties of the identified tools; C-comparison of psychometric properties, instrument by instrument; O-return a GRADE of quality of instruments and recommending those with a higher GRADE score; s-tools development or evaluation. The search steps were performed according to the PRISMA statement [19]. 

The search strategy used the key elements of the construct of interest defined with the PICOs, as well as the search filters suggested by Terwee and colleagues [20], combining them with the AND, OR, and NOT operators. An example of the query used on PubMed is given in Appendix A. To manage the research process, we used EndNote ver. 8.2. 

### 2.2. Inclusion and Exclusion Criteria

We included in this review articles fulfilling the following criteria: (1) articles on the development and/or psychometric validation of tools assessing healthcare professional’s compliance with standard precautions; (2) articles on the cultural adaptation or linguistic validation of the instrument in another country; (3) articles published in academic and peer-reviewed journals; (4) articles written in English, Italian, Peruvian, Spanish, Portuguese, and French. No limited time span was applied.

Studies that did not have as their main objective the assessment of measurement properties of the instruments that evaluated the compliance to the standard precautions (e.g., cross-sectional studies that used ad hoc surveys just to measure compliance without assessing psychometric properties) were excluded. In addition, articles that internally did not publish the tool were excluded because they needed to be evaluated by reviewers as stated in the COSMIN guidelines.

### 2.3. Qualitative Evaluation of Studies, Psychometric Properties, and Synthesis of Evidence for the Instruments

To conduct the data synthesis, we used the COSMIN guidelines. In recent years, these guidelines, initially developed to conduct systematic reviews of patient-reported outcome measures (PROMs), have been used to assess outcomes in healthy individuals or caregivers [21]. The evaluation and synthesis process is divided into two phases: one that evaluates and summarizes the evidence on the development and validation studies’ quality and one that evaluates and summarizes the evidence on the measurement properties of the instruments.

The first phase is divided into four steps. In the first step, two reviewers independently assessed the methodological quality of each study through COSMIN Box 1, which examines the relevance of the new tool’s items and the comprehensiveness and comprehensibility of the cognitive interview or the pilot study. The second step, using the COSMIN Box 2, evaluated the quality of the validation studies. This box is divided into five sections which examine relevance, comprehensiveness, and comprehensibility. Here, you can choose the sections to complete based on what was performed in the study (e.g., if the professional has not been consulted in the content validity study, sections 2d and 2e of the COSMIN Box can be skipped). In the third step, the evidence of the studies was summarized, and the tools were evaluated to determine an overall score on relevance, comprehensiveness, comprehensibility, and content validity (from sufficient to indeterminate). In the fourth step, confidence in the reliability of the overall scores (high, moderate, low, or very low) is determined using the modified Grading of Recommendations Assessment, Development, and Evaluation (GRADE) approach. According to the COSMIN 2018 guidelines, a level C is assigned when high-quality evidence for an insufficient measurement property is present, a level A rating is assigned when there is evidence for sufficient content validity and low-quality evidence for sufficient internal consistency, and a level B is assigned when the scale cannot be classified as level C or A.

The second phase is divided into a 3-steps process. In the first phase, with the COSMIN Risk of Bias checklist, two reviewers independently evaluated the methodological quality of each study. In the second phase, each measurement property has been evaluated according to the COSMIN checklist criteria. In step three, the evidence for each instrument was summarized with a rating on its psychometric properties (from sufficient to indeterminate) and the quality of the evidence (high, moderate, low, very low) using the GRADE approach. According to this approach, recommendations can be made on the use of tools: recommended for use (A level), potentially recommended but requiring further study (B level), and not recommended for use (C level).

In order to evaluate the validity of the contents and the psychometric properties, the review team used the Excel file downloadable from the COSMIN website.

### 2.4. Data Extraction

Two researchers (ML and DI), during the evaluation process, extracted from the studies some data inherent to instrument title, author, year and country of publication of the study, type of study (development or validity study), sample characteristics, number of items, response system, and psychometric properties investigated. These data were used by the review team to describe the characteristics of the studies and the psychometric characteristics of the instruments (see Table 1).

## 3. Results

### 3.1. Results of the Studies Includes in the Review

A total of 28 articles (12 on development and 16 on validation) containing 13 measurement tools were included in the review (see Figure 1). 

These studies were conducted in different continents: Asia (Hong Kong 4 studies, Iran 2 studies, China 2 studies, Turkey 2 studies, Saudi Arabia 1 study), Europe (Italy 1 study, France 1 study), Oceania (Australia 3 studies) and America (Brazil 8 studies, Bogotà 1 study, Ohio 1 study, Minneapolis 1 study, New York 1 study) (Table 1). 

**Table 1 healthcare-11-01408-t001:** Studies included in the review and psychometric properties of the instruments evaluated.

Tools	Author/ Year Publication/Country/Type of Study/Concept Assessment	Sample	Items Number/Subscale/Response System	Structural Validity	Internal Consistency	Other Psychometrics Properties
**CUPs**	Gershon et al., 1995 [22] Ohio Validation Study Compliance with UP	1716 healthcare workers (physicians, nurses, technicians, phlebotomists)	11 items for Compliance with UPs (CUPs) Fields: disposal of sharps, use of needles and barrier-protection, use of gloves, eye protection, protective outer clothing, eating or drinking in potentially contaminated areas. 5-point Likert (from 1 “Never” to 5 “Always”) 14 items for psychosocial factors (PF) 13 items for organizational management factor (OMF)		Total 0.65 (CUPs) Total 0.83 (PF) Total 0.88 (OMF)	
**CUPs**	Brevidelli & Cianciarullo, 2009 [23] Brazil Validation study Adherence with SP	270 healthcare workers (physicians and nurses)	11 items for Compliance with UPs (CUPs) Fields: disposal of sharps, use of needles, and barrier-protection such as gloves, eye protection, protective outer clothing, eating or drinking in potentially contaminated areas. 5-point Likert (from 1 “Never” to 5 “Always”) Dejoy scale	PCA (oblique and orthogonal rotations) 7 factors solution; 54.9% variance explained PCA for safety climate (OMF), 2 factors solution, 47.57% variance explained pretest (a small sample of healthcare workers)	Subdimensions 0.67–0.86 (PCA 7 factors) Subdimensions 0.69–0.80 (PCA 2 factors)	
**HAI**	O’Boyle et al., 2001 [24] Minneapolis Development study Motivation and factor of the HB	100 nurses	46 items 6 Subscales: Belief about outcomes, attitudes, referent beliefs, subjective norm, control beliefs, perceived control 7-point Likert (from 1 “Extremely unlikely” to 7 “Extremely likely”)	Face validity, 20 nurse students (pilot testing)	Subscale: 0.64–0.91	
**HAI**	Villamizar Gomez & Sànchez Pedraza, 2014 [25] Bogotà Validation study Motivation and factor of the HB	300 nurses	46 items 6 Subscales: Belief about outcomes, attitudes, referent beliefs, subjective norm, control beliefs, perceived control 7-point Likert (from 1 “Extremely unlikely” to 7 “Extremely likely”)	Face validity (6 nurses with experience in infections; comprehensiveness) EFA (varimax and promax rotation), 8 factors solution, 57.11% variance explained	Total 0.82 Subscale 0.44–0.90	Cross-cultural validity (forward and backward translation) Test-retest (recall period 5.7 weeks); r < 0.50, except hand condition= 0.54) Criterion validity (HAI and Attitudes Regarding Practice Guidelines), r < 0.30
**UPs**	Chan et al., 2002 [26] Hong Kong Development study Knowledge and compliance with UP	450 nurses	26 items Two scales: nurses’ knowledge (11 items) and nurses’ compliance (15 items) Fields: use of protective devices, disposal of sharps, disposal of waste, decontamination of spills and used articles, prevention of cross infection from person to person, contact with body fluids including tears, sweat, saliva, urine, and feces. True/false for nurses’ knowledge 4-point Likert for nurses’ compliance (from 0 “never” to 3 “always”)	Content validity (panel with 8 experts of the infection control unit, CVI = 88.6%)	Total 0.72	
**UPs**	Lam et al., 2012 [27] Hong Kong Validation study Compliance with UP	440 nurses	15 items Only nurses’ compliance UPs Fields: use of protective devices, disposal of sharps, disposal of waste, decontamination of spills and used articles, and prevention of cross-infection from person to person 4-point Likert (from 0 “never” to 3 “always”)	Face validity (15 nursing students; 100% understandability)	Total 0.80	Reliability (recall period 2 weeks): ICC: r = 0.83, *p* < 0.001, 95% CI 0.77–0.87 Hypotheses testing (known-groups technique): students and nurses, 60.2 and 69.5, respectively, t = −9.00, *p* < 0.001
**ARPG**	Larson, 2004 [28] New York Development study General and specifically attitude with Hand Hygiene Guideline	10 physicians and 11 nurses	36 items Subscales: general attitudes guideline (18 items), specifically attitudes with Hand Hygiene Guideline (18 items) 6-point Likert (from 1 “Strongly disagree” to 6 “Strongly agree”)	Content validity (panels with more than 12 experts; readability, understandability, ease of response)	Total 0.80	Test-retest for Part 1 (recall period 2 weeks); r = 0.80
**KPUPs**	Motamed et al., 2006 [29] Iran Development study Knowledge and practice with UP	540 healthcare workers and medical students	18 items 2 Subscales: Knowledge (10 items) and practices (8 items) Fields: Understanding of precautions, disposal of sharps, contact with vaginal fluid, handwashing, disposal of needles, and glove, mask, and gown usage Dichotomous response (True/False for Knowledge and Agree/Disagree for Practice subscales)	Content and face validity (panel experts of the infection control committee of the two hospitals) Face validity (pilot testing with 20 subjects, feasibility, and internal consistency)	Total 0.71	
**KAPs**	Chan et al., 2008 [30] China Development study Knowledge, attitudes and practice of operating room staff with SP and TBP	113 nurses and non-medical staff	25 items 3 subscales: Knowledge (4 items), Attitudes (11 items) and Practices (10 items) Fields: PPE, solid waste disposal, environmental cleaning and disinfection measures and safety measures following occupational exposure to biological material For knowledge scale: multiple-choice questions For attitude subscale: 5-point Likert (from 1 “Strongly disagree” to 5 “Strongly agree”) For Practices subscales: 5-point scale (from 1 “Never” to 5 “Always”)	Content validity (panel of two experts, CVI 0.97) EFA, 3 factors solution, 62.4% variance explained	Subdimension 0.71–0.89	Test-retest (recall period 2 weeks; 14 subject); r = 0.80 Hypotheses testing for construct validity (convergent validity) between attitudes and practices (r = 0.39, *p* < 0.05)
**HHQ**	Van de Mortel, 2009 [31] Australia Development study Knowledge, beliefs and practices in HH	59 student nurses	36 items Subscales: Hand hygiene belief scale (HBS, 19 items), Hand Hygiene Practices Inventory (HHPI, 14 items), Hand Hygiene Importance Scale (HIS, 3 items) Multiple choice per HBS 5-point Likert for HHPI and HIS (from 1 “Strongly disagree” to 5 “Strongly agree”)	Face validity (pilot testing with 14 nursing students; comprehension and redundancy)	Subscale 0.74–0.80	Test-retest (recall period 2 weeks; 14 subject); r > 0.79 for each subscale)
**HHQ**	Najafi Ghezeljeh et al., 2015 [32] Iran Validation study Knowledge, beliefs and practices in HH	60 student nurses	36 items Scales: Hand hygiene belief scale (HBS, 19 items), Hand Hygiene Practices Inventory (HHPI, 14 items), Hand Hygiene Importance Scale (HIS, 3 items) Multiple choice per HBS 6-point Likert for HHPI and HIS (from 0 “Never” to 5 “Always”)	Content validity (panel of 10 experts, comprehensibility) Face validity (20 student nurses, comprehensibility)	Total 0.80 Subscales 0.70–0.90	Cross-cultural validity (forward and backward translation) Test-retest (recall period 7–10 days); r > 0.51 (for each subscale); from 0.51 to 0.61 Reliability: ICC: HIS = 0.78 (0.63–0.87), HBS = 0–70 (0.60–0.81), HHPI = 0.85 (0.73–0.91)
**HHQ**	Birgili et al., 2019 [33] Turkey Validation study Knowledge, beliefs and practices in HH	595 nursing and physiotherapy students	36 items Background theory: Social cognitive theory of Bandura Subscales: Hand hygiene belief scale (HBS, 19 items), Hand Hygiene Practices Inventory (HHPI, 14 items), Hand Hygiene Importance Scale (HIS, 3 items) Multiple choice per HBS 5-point Likert for HHPI and HIS (from 1 “strongly disagree” to 5 “strongly agree”)	Content and face validity (panel of experts, comprehensibility, CVI = 0.80; CVI from 0.77 to 0.86) Face validity (15 nursing student and academic staff, comprehensibility) CFA: 5 factor solution CFI 0.97 RMSEA 0.064	Total 0.88 Subscales 0.74–0.95	Cross Cultural Validity (Forward and backward translation) Test-retest (recall period 2 weeks); r > 0.51 (for each subscale); from 0.51 to 0.61 Hypotheses testing for construct validity (convergent validity) between each pair of subscales (from 0.450 to 0.547; *p* < 0.001)
**QKCSP**	Luo et al., 2010 [34] China Development study Knowledge and compliance with SP	1444 nurses	40 items Two scales: knowledge SPs (QKSP) and compliance SPs (QASP) Fields: Hand hygiene, PPE, safe handling of patient care equipment, safe practices in the handling of piercing and cutting objects and safety measures following occupational exposure to biological material 3-point Likert for Knowledge SPs (yes, no and uncertain/unknown) 4-point Likert for Compliance SPs (from 0 “Never” to 4 “Always”)	Content and face validity QKSP (0.98) Content and face validity QASP (0.98)	Total QKSP 0.92 Total QASP 0.93	Test-retest QKSP (recall period not indicated); r = 0.86 Test-retest QASP (recall period not indicated); r = 0.87
**QKCSP**	Valim et al., 2013 [35] Brazil Validation Study Knowledge and compliance with SP	42 nurses	40 items Two scales: knowledge SPs (QKSP) and compliance SPs (QASP) Fields: Hand hygiene, PPE, safe handling of patient care equipment, safe practices in the handling of piercing and cutting objects and safety measures following occupational exposure to biological material 3-point Likert for Knowledge SPs (yes, no and uncertain/unknown) 4-point Likert for Compliance SPs (from 0 “Never” to 4 “Always”)	Content validity (panel of 12 experts; semantic evaluation) Face validity (30 nurses; understand and clarity)		Cross-cultural validity (forward and backward translation)
**CSPS**	Lam, 2011 [36] Hong Kong Development study Compliance with SP	193 nurses	20 items Fields: Use protective devices, disposal and sharp, disposal of waste, decontamination, prevention cross infection 4-point Likert (from 1 “Never” to 4 “Always”)	Content validity (panel with six experts on the infection theme, relevance, and adequacy, CVI = 0.90, CVI-item = 0.83–1.00) Face validity (72 nurses and nursing students, 100% understandable words and style)	Total 0.73	
**CSPS**	Lam, 2014 [37] Hong Kong Validation study Compliance with SP	453 nurse and nursing students	20 items Fields: Use protective devices, disposal and sharp, disposal of waste, decontamination, prevention cross infection 4-point Likert (from 1 “Never” to 4 “Always”)	Reviewed by 19 international experts with narrative feedback (relevance and globally applicable)	Total 0.73	Reliability (recall period 2 weeks and 3 months): ICC: r = 0.79, *p* < 0.001 (2 weeks) ICC: r = 0.74, *p* < 0.001 (3 months) Criterion validity (CSPS and UPs), r = 0.76, *p* < 0.001
**CSPS-A** **Arabic version**	Cruz et al., 2016 [38] Saudi Arabia Validation study Compliance with SP	230 nurses	20 items Fields: Use protective devices, disposal and sharp, disposal of waste, decontamination, prevention cross infection 4-point Likert (from 1 “Never” to 4 “Always”)	Content validity (panel of 5 experts in infection control; relevance; CVI = 1) Pilot-testing (40 nurses, difficult and understand)	Total 0.89	Cross-cultural validity (forward and backward translation) Reliability (recall period 2 weeks): ICC = 0.88 Hypotheses testing (known-groups technique): - the nursing staff had higher SP compliance than nursing students - positive correlations between students’ clinical experience and SP compliance (r = 0.48, *p* < 0.001)
**CSPS-PB** **Portughese-Brasilian version**	Pereira et al., 2017 [39] Brazil Validation study Compliance with SP	300 nurses	20 items Fields: Use protective devices, disposal and sharp, disposal of waste, decontamination, prevention cross infection 4-point Likert (from 1 “Never” to 4 “Always”)	Content validity (panel of 7 experts; relevance, comprehensiveness, comprehensibility) Pilot-test (50 nurses, comprehensibility)	Total 0.61	Cross-cultural validity (forward and backward translation) Reliability (recall period 2 weeks): ICC = 0.85, *p* < 0.001
**CSPS-It** **Italian version**	Donati et al., 2019 [3] Italy Validation study Compliance with SP	253 nurses	20 items Fields: Use protective devices, disposal and sharp, disposal of waste, decontamination, prevention cross infection 4-point Likert (from 1 “Never” to 4 “Always”)	Content validity (panel of 6 experts; relevance, comprehensiveness, comprehensibility, CVI = 0.95) CFA (unidimensional model) CFI = 0.90 TLI = 0.87 RMSEA = 0.09	Total 0.84	Cross-cultural validity (Forward and backward translation) Reliability (recall period 2 weeks): ICC = 0.86, *p* < 0.001 Hypotheses testing (known-groups technique): compliance of nurses who attended a training course on SPs was significantly higher (*p* < 0.001)
**CSPS-T** **Turkey version**	Samur et al., 2020 [5] Turkey Validation study Compliance with SP	411 nurses	20 items Fields: Use protective devices, disposal and sharp, disposal of waste, decontamination, prevention cross infection 4-point Likert (from 1 “Never” to 4 “Always”)		Total 0.71	Cross-cultural validity (forward and backward translation) Reliability (recall period 2 weeks): ICC = 0.84, CI 95% 0.77–0.90, *p* < 0.001
**QCSP**	Valim et al., 2015 [40] Brasil Validation study Compliance with SP	121 nurses	20 items Fields: Hand hygiene, protective equipment (gloves, mask, goggles, and apron) and disposable equipment (hat and shoes), use and disposal of needles, blades, and sharps in specific containers, the procedure in the case of injuries from potentially contaminated sharps 5-point Likert (from 0 “never” to 4 “always”)		Total 0.80	Reliability (recall period 2 weeks): ICC: r = 0.973, 95% CI 0.93–0.99, *p* < 0.001 Hypotheses testing for construct validity (convergent validity) between compliance to SP and higher nurses’ perceived safety (r = 0.614; *p* < 0.001) Hypotheses testing for construct validity (discriminant validity) between compliance to SP and perception of obstacles to follow precautions (r = −0.537; *p* < 0.001)
**SPQ**	Michinov et al., 2016 [41] France Development study Compliance with SP	331 healthcare workers (nurses, physicians, and medical students)	24 items 7 Subdimension: Attitude, social influence, facilitating organization, exemplary behavior, organizational constraint, individual constraint, intention Fields: prevention of infection, influence and exemplary behavior of colleagues, facilities available in a health care setting, training and reminders in the use of SP, the occurrence of unanticipated events, lack of time, heavy workload, lack of knowledge about SP, personal beliefs, problems related to use of equipment 5-point Likert (format not indicated)	Face validity (panel of 5 experts; understand and clarity of items) Face validity (14 nurses; reformulation and redundant items) EFA 7 factors solution, 66.51% variance explained	Total 0.78 Subdimension 0.71–0.88	
**SPQ**	Pereira-Avila et al., 2019 [42] Brazil Validation study Compliance SP	21 healthcare workers (physicians and nurses)	24 items 7 Subdimension: Attitude, social influence, facilitating organization, exemplary behavior, organizational constraint, individual constraint, intention Fields: prevention of infection, influence and exemplary behavior of colleagues, facilities available in a health care setting, training and reminders in the use of SP, the occurrence of unanticipated events, lack of time, heavy workload, lack of knowledge about SP, personal beliefs, problems related to use of equipment 5-point Likert (format not indicated)	Content and face validity (panel of 5 experts; clarity, understanding, and relevance, CVI 0.96) Semantic evaluation (21 healthcare workers)		Cross-cultural validity (forward and backward translation)
**SPQ**	Luna et al., 2020 [43] Brazil Validation study Compliance SP	300 healthcare workers (physicians and nurses)	24 items 7 Subdimension: Attitude, social influence, facilitating organization, exemplary behavior, organizational constraint, individual constraint, intention Fields: prevention of infection, influence and exemplary behavior of colleagues, facilities available in a health care setting, training and reminders in the use of SP, the occurrence of unanticipated events, lack of time, heavy workload, lack of knowledge about SP, personal beliefs, problems related to use of equipment 5-point Likert (format not indicated)	EFA (varimax rotation), 7 factors solution; 65.75% variance explained	Total 0.71 Subdimension 0.69–0.83	Hypotheses testing (known-groups technique): nurses significantly correlation with intention (*p* = 0.000) and individual constraint (*p* = 0.041) respect physicians and nursing technicians
**FIASP**	Bouchoucha & Moore, 2019 [44] Australia Development study Adherence SP	684 nurses	29 items Subdimensions: Leadership, justification, contextual cues, culture/practice, judgment 5-point Likert (from 0 “Not at all” to 4 “Very much”)	PCA (oblique rotation), 5 factors solution, 48% variance explained CFA, 5 factors solution GFI = 0.889 RMSEA = 0.038 SRMR = 0.054	Subdimension 0.61–0.85	Reliability (recall period 4 weeks): ICC: range r = 0.69–0.85, *p* < 0.001
**FIASP**	Bouchoucha et al., 2021 [45] Australia Validation study Adherence SP	321 undergraduate nursing students	29 items Subdimensions: Leadership, justification, contextual cues, culture/practice, judgment 5-point Likert (from 0 “Not at all” to 4 “Very much”)	Face validity (panel of 6 experts; comprehensiveness) PCA (oblique rotation), 4 factors solution, 53.82% variance explained CFA, 4 factors solution CFI = 0.89 RMSEA = 0.05 SRMR = 0.08	Subdimension 0.79–0.80	
**AGHPC**	Meneguin et al., 2022 [46] Brazil Development study Adherence Good Practices for COVID-19	35 healthcare workers	47 items 3 subdimensions: personal, organizational, and psychosocial 5-point Likert (from 1 “Never” to 5 “Always”)	Content and face validity (panel of 7 experts; clarity, relevance, and comprehensiveness; CVI 0.99) Face validity (35 healthcare workers; understanding)		
**AGHPC**	Meneguin et al., 2022 [47] Brazil Development study Adherence Good Practices for COVID-19	307 healthcare workers	47 items 3 subdimensions: personal, organizational, and psychosocial 5-point Likert (from 1 “Never” to 5 “Always”)	EFA (oblique rotation) 3 factors solution; 78.2% variance explained CFA, 3 factors solution CFI = 0.996 TLI = 0.995 RMSEA = 0.072 SRMR = 0.082	Total 0.96 Subdimension 0.61–0.95	Hypotheses testing for construct validity (convergent validity) between total score and its domains (r 0.66–0.90; *p* < 0.001)

**Note**: UP = Universal Precautions; HB = handwashing behavior; SP = Standards Precautions; TBP = transmission-based precautions; EFA = exploratory factor analysis; PCA = principal component factor analysis; CFA = confirmatory factor analysis; CUPs = Compliance with Universal Precautions; HAI = Handwashing Assessment Inventory; KPUPs = Knowledge and Practices Universal Precautions Scale; KAP = Knowledge, Attitudes and Practices scale; HHQ = Hand Hygiene Questionnaire; UPs = Universal Precaution scale; ARPG = Attitudes Regarding Practice Guidelines; QKCSP = questionnaires for knowledge and compliance with standard precaution; CSPS = Compliance with Standard Precautions Scale; QCSP = questionnaire for compliance with standard precaution; SPQ = Standard Precautions Questionnaire; FIASP = Factors Influencing Adherence to Standard Precautions Scale; AGHPC = Adherence to Good Hospital Practices for COVID-19.

The tools identified can be classified into four categories. The first category includes tools that assess adherence to Universal Precautions (UPs): Compliance with Universal Precautions (CUPs) [22,23], Knowledge and Practices Universal Precautions Scale (KPUPs) [29], and Universal Precaution scale (UPs) [26,27].

The second category includes tools that assess compliance with Standard Precautions (SPs): Questionnaires for Knowledge and Compliance with Standard Precaution (QKCSP) [25,31], Compliance with Standard Precautions Scale (CSPS) [3,5,36,37,38,39], Questionnaire for Compliance with Standard Precaution (QCSP) [40], Standard Precautions Questionnaire (SPQ) [41,42,43], Factors Influencing Adherence to Standard Precautions Scale (FIASP) [44,45].

The third category includes instruments that assess attitudes, behaviors, and beliefs that affect hand hygiene adherence (HH): Handwashing Assessment Inventory (HAI) [24,25] and Hand Hygiene Questionnaire (HHQ) [31,32,33].

Finally, the fourth category includes tools that assess compliance with guidelines and Transmission-based precautions (TBPs): Knowledge, Attitudes and Practices scale (KAPs) [30], Attitudes Regarding Practice Guidelines (ARPG) [28], and Adherence to Good Hospital Practices for COVID-19 (AGHPC) [46,47].

The descriptions of the studies and the instruments, with their psychometric properties, are presented in Table 1.

### 3.2. Methodological Quality, Overall Rating, and GRADE’s Quality of Evidence

In the evaluation of the quality of evidence, seven instruments were rated Moderate (CSPS, FIASP, HAI, HHQ, QKCSP, SPQ, UPs), 1 Low (AGHPC), and 5 Very Low (ARPG, CUPs, KAPs, KPUPs, QCSP). This was determined by the quality and quantity of the validation and development studies reviewed.

Despite the low scores obtained (low and very low), the studies were not excluded from subsequent evaluations in accordance with the COSMIN guideline.

Contributing to the low scores for relevance, completeness, comprehensibility, and content validity were some biases in the design of the studies, which received mostly dubious ratings. Such ratings were assigned mainly to the instrument development procedures and pilot tests. In fact, many studies did not give clear and comprehensive information about the qualitative methodology for identifying relevant items, the presence of trained moderators or interviewers, the publication of interview guidelines in the article, the process of recording and transcribing participants’ responses, the process of independent data coding, and the achievement of qualitative data saturation.

In addition, in the pilot tests, clear and comprehensive statements on the relevance, completeness, and comprehensibility of the items were not provided by the respondents. Often also, the number of people enrolled in the pilot test was as insufficient as those included in the expert panel, where sometimes it was not specified what expertise they had. In addition, for some instruments (ARPGs, KPUPs, and KAPs), only developmental studies with questionable quality ratings were included in the review, further penalizing the GRADE rating. For the GRADE evidence quality scores, see Table 2.

### 3.3. Psychometric Properties, Overall Rating and GRADE’s Quality of the Evidence

At the stage of assessing the psychometric properties of the instruments included in the review, six instruments were rated of moderate quality (CSPS, CUPs, FIASP, HHQ, KPUPs, and SPQ) and seven instruments as low (AGHPC, ARPG, HAI, KAPs, QCSP, QKCSP, and UPs).

These scores were determined by the procedures used to test psychometric properties and were influenced by some biases. For example, low scores were assigned if, in the structural validity test, the sample size was not adequate for analysis (adequate rating: at least 5 times the number of items and ≥100 or 6 times the number of items, but <100). 

Based on the psychometric properties analyzed in the studies and shown in Table 1, we were able to assess whether they met the criteria of good measurement properties given in the COSMIN guidelines.

Finally, based on the quality of the studies and the good psychometric properties of the instruments, we provided recommendations according to the modified GRADE method given in the COSMIN guidelines. An instrument that scored GRADE A had sufficient content validity (+) at any level of evidence and at least low-quality evidence for sufficient internal consistency. An instrument that scored GRADE C had to have high-quality evidence for an insufficient measurement property. GRADE B was assigned in cases GRADE A or C was not assigned. However, we considered assigning GRADE C to instruments that had been less recently developed and not further validated, had inconsistent content validity, and insufficient psychometric properties of at least a moderate degree.

Finally, four instruments were given a GRADE A rating (SPQ, HHQ, QCSP, and CSPS), seven instruments a B rating (AGHPC, ARPG, FIASP, KAPs, QKCSP, UPs, and HAI), and two a C rating (CUPs and KPUPs).

### 3.4. Compliance Standard Precaution Instruments

There were 13 instruments included in this review; we present here a brief narrative overview of the instruments. For a complete overview of the instruments and the procedures adopted in their development and validation, see Table 1.

The Compliance with Universal Precautions scale (CUPs) originates from the Work System Model of Dejoy and colleagues [22,23], which states that compliance with universal precautions occurs at three levels: health care worker, work dynamics, and organizational context. It is a scale consisting of 11 items with a 5-point Likert response system. The total scale score ranges from 11 to 55, and high scores represent high levels of compliance with UPs. The items assessed the frequency with which workers followed specific recommendations during their work, such as, for example, proper disposal of sharps and needles, use of barrier protection (gloves, eye protection, protective clothing), and poor habits such as eating or drinking in potentially contaminated areas. Two validation studies [22,23] were included in the review; the first assessed compliance with UPs [22], and the second assessed adherence to SPs. Both used CUPs internally along with other instruments and assessed their psychometric properties. These two studies had very low and inconsistent content validity (±/VL) because the procedures for comprehensiveness and comprehensibility were not clearly described, which reported inconsistent or indeterminate ratings. Finally, the GRADE rating of this instrument is C, having also obtained an insufficient score on the psychometric properties measured.

The Handwashing Assessment Inventory (HAI) was developed by O′Boyle and colleagues in 2001 [24] to assess factors that motivate health care workers to handwash. The scale is based on the Theory of Planned Behavior (TPB) [48], which assesses how individuals modify their behavior, which, in this case, consists of performing and complying with proper handwashing. The scale consists of 46 items divided into 6 sections representing the 6 principles of planned behavior in hand hygiene: beliefs about outcomes, attitudes, referent beliefs, subjective norms, control beliefs, and perceived control. The response system is a 7-point Likert. The score for each of the sections is calculated by summing the individual item scores and dividing by the number of items each participant responded to. Negatively worded items should be poured into the score before calculating the scores for each section of the HAI. Higher scores in the HAI reflect more positive motivation in hand washing. The scale development study [24] and the validation study conducted in a hospital in Bogota [25] were included in the review. The GRADE assessment of this instrument was type B because there was low evidence of insufficient psychometric properties and inconsistent content validity of a moderate degree.

The Universal Precaution scale (UPs) was developed by Chan and colleagues in 2002 [26] and was used to assess compliance with UPs by nursing students and nurses [26,27]. The items were constructed based on the UPs guideline recommended by the Hong Kong Hospital Authority in November 1988. The questionnaire consists of three parts. The first collects demographic data. The second part assesses knowledge about universal precautions and consists of 11 items with a dichotomous response system (true or false). A score of 1 is assigned to each correct answer, and the maximum possible score is 11. The higher the score, the higher the knowledge about UPs. The third part assesses compliance with universal precautions. It consists of 15 items with a 4-point Likert response system. The total scale score ranges from 0 to 33. The higher the score, the higher the compliance with UPs. The areas explored by the instrument are the use of protective equipment, sharps disposal, disposal of contaminated waste, decontamination of patient body fluids and instruments used in care, and prevention of person-to-person cross-infection. There were two studies included in the review, one developmental [26] and one validation [27]. The GRADE of the instrument is type B because the content validity is inconsistent and of moderate grade, and the evidence on psychometric properties is sufficient but of low grade. 

The Attitudes Regarding Practice Guidelines (ARPG) was developed in 2004 [28] to assess obstacles to adherence to the guidelines in general and to hand hygiene. There are 18 items for the general part and 18 items for the specific part. The response system is a 6-point Likert scale. Possible scores for the two subscales range from 0 to 108, with higher scores indicating fewer perceived obstacles. In addition, the instrument asks the respondent to indicate the most important factors that facilitated or hindered guideline use and to self-report the percentage of times he or she uses a hand alcohol product. The development study, where the methodological quality was doubtful, was included in the review [28]. No other validation studies of the instrument were found. The instrument’s content validity obtained an inconsistent and very low score because the procedure for attesting by experts the comprehensibility and relevance of the items were not clearly stated, and the procedure for the target group of interest was not declared. In spite of this, the GRADE of the instrument is type B for a sufficient, although low psychometric property.

The Knowledge and Practices Universal Precautions Scale (KPUPs) was developed in 2006 [29] to assess the knowledge and practice of UPs, based on the guidelines recommended by the CDC in 1987 (12 items) [49] and a questionnaire devised by Chan et al. in Hong Kong [26]. Consisting of 18 items, it is divided into two subscales: knowledge (10 items) and practices (8 items). The response system is dichotomous (True/False for Knowledge and Agree/Disagree for Practice subscales). The maximum score for knowledge is 10, and the higher it is, the higher the knowledge about UPs. The maximum score for practices is 8, and the higher it is, the higher the adherence in practice to UPs. The fields investigated are the understanding of precautions, disposal of sharps, contact with vaginal fluid, handwashing, disposal of needles and gloves, and mask and gown usage. The GRADE of the instrument is C because the content validity is inconsistent and very low, and the psychometric properties are insufficient with a moderate level of evidence.

The Knowledge, Attitudes and Practices scale (KAPs) was developed in 2008 [30] based on the literature review and expert opinion. The instrument includes demographic questions and a series of items that form subscales. The first subscale is on knowledge of SPs and TBPs with four multiple-choice questions. Correct answers receive one point; the total score therefore ranges from 0 to 4, and higher scores indicate a higher knowledge level. The second subscale consists of 11 items assessing attitudes in choosing Personal Protective Equipments (PPEs) (three items), wearing PPEs (four items), and handling high-risk procedures (four items). The response system is a five-point scale, and higher scores indicate stronger agreement on attitudes. The third subscale consists of 10 items on practices in wearing gloves (three items), wearing gowns and eye shields/goggles (four items), and following the precautionary guidelines and the contingency plan (three items). The response system is a five-point scale, and higher scores indicate greater compliance. The instrument earned a GRADE of type B because it had consistent content validity of very low grade but sufficient internal consistency of low grade.

The Hand Hygiene Questionnaire (HHQ) was developed in 2009 [31] to assess knowledge, beliefs, and practices in hand washing. It has Bandura’s social cognitive theory [50] as its guiding theory. The HHQ includes three scales (36 items): a hand hygiene beliefs scale (HBS) (19 items), a hand hygiene importance scale (HIS) (3 items), and a hand hygiene practices inventory (HHPI) (14 items). The response system is a multiple choice one for HBS and a 5-point Likert one for HHPI and HIS. Three studies were included in the review, one developmental [31] and two validation [32,33]. The ratings of the studies were sufficient and moderate for relevance, comprehensiveness, comprehensibility, and overall content validity ratings. The psychometric properties of the instrument were sufficient and moderate. Therefore, the GRADE obtained from the instrument was type A.

The Questionnaires for knowledge and compliance with standard precaution (QKCSP) were developed in 2010 in China [34]. Its development originates from the guidelines of the Centers for Disease Control and Prevention (CDC) in the United States, which established the concept of standard precautions (SP) in 1996 [51]. The SP knowledge questionnaire includes 20 questions, with possible “yes”, “no”, or “don’t know” answers. One point is added for each “yes”, and the maximum possible score is 20 points. The higher the score, the greater the knowledge. The SP adherence questionnaire includes 20 questions with a 4-point Likert response system. The total possible score ranges from 0 to 80 points. The higher the score, the greater the individual’s adherence to SPS. A developmental study [34] and a validation study [35] were included in the review. In the development study, unlike the validation study, the procedures that assessed the face validity of the instrument were not clearly described, producing indeterminate and moderate results in content validity (±/M). THE GRADE of the instrument is type B because the internal consistency was sufficient although low (+/L).

The Compliance with Standard Precautions Scale (CSPS) was developed in 2011 [36] by modifying the 15 items of Chan and colleagues’ 2002 UPs and adding others. The final scale consists of 20 items, with a 4-point Likert response system to assess the use of protective equipment, disposal of sharps and waste, decontamination from biological fluids of used instruments, and prevention of cross-infection. Higher values indicate better compliance with SPs. The CSPS has been translated into several languages and adopted in various countries, including, but not limited to, Arabic [38], Portuguese-Brazilian [39], Italian [3], and Turkish [5]. One developmental study [36] and five validation studies that met the id inclusion citers of the review [3,5,37,38,39] were included in the review. The scale achieved a GRADE of type A because content validity was sufficient and of moderate quality (+/M), and internal consistency was sufficient and of moderate quality (+/M).

The Questionnaire for compliance with standard precaution (QCSP) was developed in 2013 by Felix in a doctoral dissertation on a sample of 1444 Chinese nurses. It consists of twenty Likert scale questions with scores from 0 to 4 points, the total score ranges from zero to eighty points, and higher scores indicate high compliance with SPs. The development study was not included in the review because it was a doctoral thesis, but only a validation study was included [40]. The methodological quality of the validation study was very low, and face and content validity were not assessed. However, the reviewers considered the items valid from a relevance, comprehensiveness, and comprehensibility point of view. Therefore, content validity was rated as sufficient, although very low. The instrument received a GRADE of type A because the psychometric properties had sufficient scores even though they were of low quality.

The Standard Precautions Questionnaire (SPQ) was developed in 2016 in France [41] for the purpose of determining socio-cognitive factors, attitudes, behaviors, limitations, and individual and organizational constraints to SPs compliance. A development article [41] and two validation studies [41,42] were included in the review. From the literature review of existing instruments and analysis of some interviews, a 35-item questionnaire was developed with a 5-point Likert response system, later reduced to 24 items and 7 subdimensions: exemplary behavior (2 items), organizational constraints (4 items), intention to perform standard precautions (4 items), social influence (4 items), attitude toward standard precautions (3 items), facilitating organization (3 items) and individual constraints (4 items). The items are visually organized into five parts: knowledge about SPs, work environment, factors that enable compliance with SPs, factors that hinder compliance with SPs, and intention to comply with SPs. The domains explored by the instrument are prevention of infection, influence and exemplary behavior of colleagues, facilities available in a health care setting, training and reminders in the use of SP, the occurrence of unanticipated events, lack of time, heavy workload, lack of knowledge about SP, personal beliefs, problems related to use of equipment. The instrument demonstrates moderate to sufficient content validity and sufficient to moderate internal consistency such that it was assigned a GRADE score of type A.

The Factors Influencing Adherence to Standard Precautions Scale (FIASP) was developed in 2019 [44] to explore factors that may impact nurses’ adherence to SPs. The FIASP scale, which in the developers’ final form consists of 29 items, measures five influential factors on nurses’ adherence to SPs that include leadership among colleagues, awareness of environmental stimuli for SPs implementation, an organization promoting or hindering SPs implementation, making professional judgments or evaluating situations and patients, and justifications nurses may give to justify their adherence or non-adherence to SPs. A development study [44] and a validation study [45] from FIASP were included in the review. The score assigned to content validity was affected by the unclear description of procedures to assess the relevance, comprehensiveness, and comprehensibility of items. However, internal consistency was rated as sufficient and of moderate quality, and the instrument was assigned a GRADE of type B.

The Adherence to Good Hospital Practices for COVID-19 (AGHPC) was developed in 2022 by Meneguin and colleagues [46,47] for the purpose of assessing the adherence of healthcare providers to good practices for COVID-19 in the hospital setting. The Health Belief Model, developed by U.S. psychologists in 1950, was used as the theoretical framework for developing the instrument [52]. According to this model, the adoption of preventive behavior depends on considering oneself vulnerable to a particular health problem that may affect us sooner or later (perceived susceptibility), perceiving that the health problem may have serious consequences (perceived severity), believing that the health problem can be prevented by a particular action (perceived benefits) regardless of whether that action involves negative aspects (perceived barriers). The AGHPC consists of 47 items with a 5-point Likert response system and 3 subdimensions: personal, organizational, and psychosocial. The instrument achieved sufficient content validity because the procedures to assess relevance, comprehensiveness, and comprehensibility were clearly and comprehensively described [46,47]. However, the quality of evidence is low because no validation studies were found in the literature but only the two developmental ones. However, the GRADE of the instrument is type B because the psychometric properties had poor ratings in both structural validity and internal consistency that were of low grade.

## 4. Discussion

In our systematic review, a total of 28 studies emerged that estimated the reliability and validity of 13 instruments assessing healthcare workers’ SPs compliance in 13 different countries belonging to 4 continents (Asia, Europe, America, and Oceania). Most of the studies were conducted in Asia and America (23 studies). The first instrument developed was the CUPs in 1995 [22], and the last was the AGHPC in 2022 [46,47]. This means that this research topic has been covered for almost 30 years, in which there have been several modifications and changes in international knowledge and guidelines for the prevention of HAIs (Healthcare Associated Infections) [26,34].

The tools identified can be classified into four categories: tools that assess adherence to Universal Precautions (CUPs, KPUPs, and UPs), instruments that assess compliance with Standard Precautions (QKCSP, QCSP, CSPS, SPQ, and FIASP), scales that assess attitudes, behaviors, and beliefs that affect hand hygiene adherence (HAI and HHQ), tools that assess compliance with guidelines and transmission-based precautions (KAPs, ARPG, and AGHPC).

In the first category, compliance with Universal Precautions was assessed [22,26,29]. The tools were designed based on national and CDC guidelines (UPs and KPUPs) and on the Work System Model of Dejoy [22]. Of these instruments, only UPs reach the recommendation level of GRADE B, while CUPs and KPUPs achieve level C. For CUPs and KPUPs, this was because in the studies included in the review, content validity and internal consistency did not achieve sufficiency, and the quality of evidence was moderate. In contrast, for UPs, internal consistency was sufficient, but content validity was indeterminate, which did not allow it to reach a GRADE level A of recommendation. 

The second category, on the other hand, includes tools designed after 2009 that assess compliance with Standard Precautions. These instruments were developed based only on the literature review (QKCSP and QCSP), on a combination of the literature review and interviews with healthcare workers (SPQ and FIASP), or on a modification of a scale that assessed UPS compliance (CSPS). Of these tools, three achieved GRADE level A because content validity and internal consistency were sufficient. Therefore, when measuring compliance with standard precautions, we recommend using QCSP, SPQ, and CSPS as instruments with higher psychometric quality.

The third category includes instruments that assess healthcare workers’ compliance with hand hygiene and originate from Bandura’s behavioral theories (HHQ) or planned behavior theories (HAI). Among these instruments, HHQ had a GRADE level A; therefore, we suggest its use when evaluating compliance with hand hygiene.

Finally, the last category included tools that assess compliance with additional precautions, such as the KAPs, which assess compliance with SPs and TBPs. The AGHPC assesses compliance with good practices for COVID-19, and the ARPG assesses obstacles to adherence to the guidelines in general and to hand hygiene. All three instruments achieved only a GRADE level B. Therefore, further research is needed to test in other settings the present instruments or to develop others of a better quality.

Common problems in the evaluation of studies included the challenging comparison of results from different studies that included the same instruments. This was due to the methodological quality adopted being heterogeneous and the validation studies being conducted at different times, where some analyses may not have been known at the time or may have become obsolete over time. Another problem encountered was that the internal consistency and structural validity estimated for most of the instruments were evaluated with methodological approaches of different quality, also compromising the quality of the evidence for the results. Finally, convergent validity and criterion validity were assessed on a few occasions, i.e., in instruments assessing SPs or TBs (CSPS, KAPs, and QCSP) or those assessing hand hygiene compliance (HAI and HHQ). We hypothesize that in other instruments, they were not assessed due to a lack of field knowledge and instruments that could represent the gold standard of comparison.

### Limitations

One of the limitations of this review may have been that it included only peer-reviewed studies in academic journals and placed language limitations. Therefore, this may have resulted in potential publication selection bias, as other tools may have been developed and disseminated as gray literature or in different languages. In any case, we tried to learn about possible tool developments when only validation studies had been found, as in the case of the QCSP. The evaluation of the studies was based on the COSMIN 2018 guidelines, and some of the criteria required for “very good” or “adequate” evaluation may not have been considered by the authors of the older studies and thus may have influenced the final evaluation of the instruments. Finally, it was not possible to assess the responsiveness of the instruments, that is, the ability of an instrument to detect a change in the measured construct over time (as required by the COSMIN procedure) due to the absence of longitudinal studies among those included.

## 5. Conclusions

Thirteen instruments assessing healthcare workers’ SPs compliance have undergone a validation process so far. Some have been developed from behavioral theories, some from literature reviews, and some have blended, revised, and integrated several already validated instruments. Not all relevant psychometric properties have been evaluated for the instruments, and often the methodological approaches used are dubious or inadequate. In addition, a lack of homogeneity in the procedures for both assessing the relevance, completeness, and comprehensibility of the instruments and assessing psychometric properties has emerged, thus threatening the external validity of the instruments. It is necessary to address future research by completing the validation processes undertaken for newly developed and already developed instruments but using higher quality methods and estimating all psychometric properties.

## Figures and Tables

**Figure 1 healthcare-11-01408-f001:**
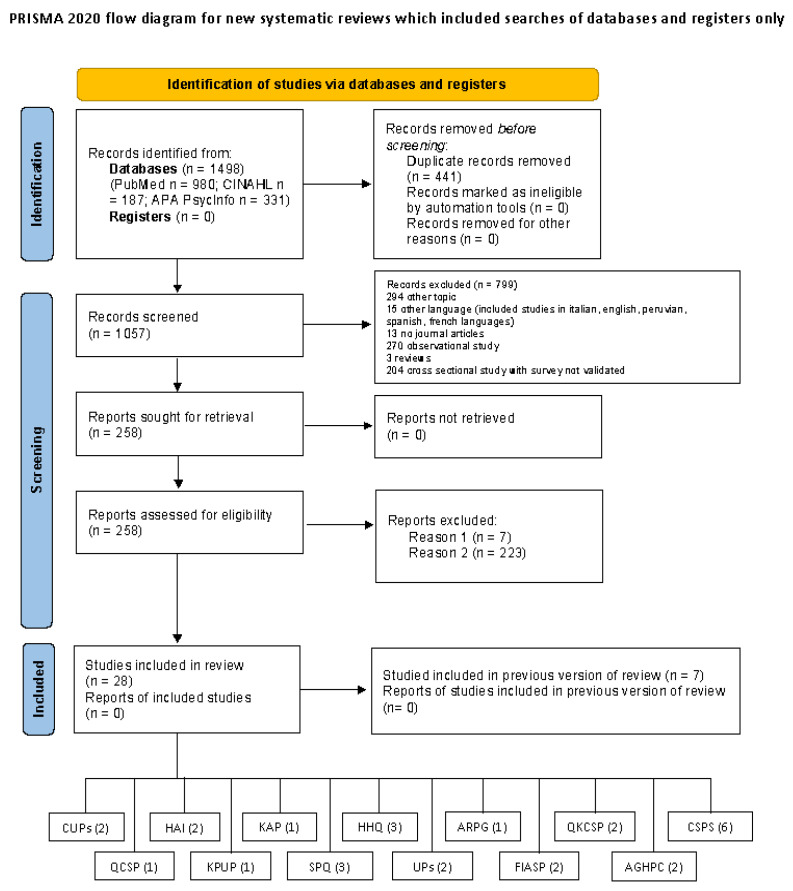
PRISMA flow diagram. Reason 1 = instruments not included in article; Reason 2 = not validation studies (e.g., survey).

**Table 2 healthcare-11-01408-t002:** Evaluation of content validity and psychometric properties and development of recommendations for the development of the instrument.

Tool	Relevance	Comprehensiveness	Comprehensibility	Overall Content Validity	Structural Validity	Internal Consistency	Other Measurement	Recommendation
**AGHPC**	+/L	+/L	+/L	+/L	−/L	−/L	Hypothesis testing +/L	**B**
**ARPG**	+/VL	±/VL	±/VL	±/VL		+/L	Reliability +/L	**B**
**CSPS**	+/M	+/M	+/M	+/M	−/M	+/M	Reliability +/M Cross-cultural validity +/M Hypothesis testing −/M	**A**
**CUPs**	+/VL	±/VL	?/VL	±/VL	-/M	-/M		**C**
**FIASP**	±/M	±/M	±/M	±/M	?/M	+/M	Reliability +/M	**B**
**HAI**	+/M	±/M	+/M	±/M	−/L	−/L	Cross-cultural validity +/L Criterion validity −/L Reliability −/L	**B**
**HHQ**	+/M	+/M	+/M	+/M	+/M	+/M	Cross-cultural validity +/M Reliability ?/M Hypothesis testing −/M	**A**
**KAPs**	+/VL	±/VL	±/VL	±/VL	−/L	+/L	Reliability +/L Hypothesis testing −/L	**B**
**KPUPs**	±/VL	±/VL	±/VL	±/VL		−/M		**C**
**QCSP**	+/VL	+/VL	+/VL	+/VL		+/L	Hypothesis testing +/L Reliability +/L	**A**
**QKCSP**	±/M	±/M	±/M	±/M		+/L	Reliability +/L Cross-cultural validity +/L	**B**
**SPQ**	+/M	+/M	+/M	+/M	−/M	+/M	Hypothesis testing +/M Cross-cultural validity +/M	**A**
**UPs**	+/M	±/M	±/M	±/M		+/L	Reliability +/L Hypothesis testing +/L	**B**

**Note**: + = sufficient; − = insufficient; ± = inconsistent; ? = indeterminate; H = High; M = Moderate; L = Low; VL = Very low; A = sufficient content validity (any level) and at least low-quality evidence for sufficient internal consistency; B = non A and non C; C = high-quality evidence for an insufficient measurement property; CUPs = Compliance with Universal Precautions; HAI = Handwashing Assessment Inventory; KPUPs = Knowledge and Practices Universal Precautions Scale; KAPs = Knowledge, Attitudes and Practices scale; HHQ = Hand Hygiene Questionnaire; UPs = Universal Precaution scale; ARPG = Attitudes Regarding Practice Guidelines; QKCSP = Questionnaires for knowledge and compliance with standard precaution; CSPS = Compliance with Standard Precautions Scale; QCSP = Questionnaire for compliance with standard precaution; SPQ = Standard Precautions Questionnaire; FIASP = Factors Influencing Adherence to Standard Precautions Scale; AGHPC = Adherence to Good Hospital Practices for COVID-19.

## Data Availability

Not applicable.

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
