# Peer review of "Evaluation of Standard Precautions Compliance Instruments: A Systematic Review Using COSMIN Methodology"

_healthcare, 2023, doi:10.3390/healthcare11101408_

Round 1

Reviewer 1 Report

Title: OK; however, it is better to sate the title in a measurable format.

Abstract: I think it is necessary to convey the objectives of the study with SMART consideration (Specific, Measurable, Attainable, Relevant and Time bound). In this case the result and the conclusion sections will be justifiable for the readers. It seems necessary that the conclusion to be supported by the evidences provided in this section.

Introduction: It is necessary to state the applied use of the expected results of the study or the novelty of the paper in this section. Please show the objectives of the study according to the PICO in this section as well.

Material and method: please notice to the following comments:

-          Please define the inclusion and exclusion criteria of the recruited studies in more detail with explicit format and according to the defined PICO. These criteria shall include the type of the design of the studies and time of publication of them.

-          Please define the variables the information of which were proposed to be extracted from the recruited studies.

Results and the discussion: because of the above-mentioned problems these sections are not justifiable.

Reviewer 2 Report

Thank you for submitting this review entitled 

 Evaluation of instruments assessing health professionals' 2 compliance with standard precautions: a systematic review

It is an interesting topic that addresses the evaluation of instruments assessing health professionals' compliance with standard precautions.

It would be necessary to place the figures and tables after naming them in the text as appropriate.

You should modify some of the information in your manuscript, which I believe is erroneous.

Line 113, the Boolean operators used in your review according to the table are AND and OR and not as you refer in the Material and Methods section.

The references do not follow the reference format of the journal, this is a corrected example, all of them should be reviewed

1.Siegel. Guideline for Isolation Precautions: Preventing Transmission of Infectious Agents in Health Care Settings - American Journal of Infection Control [Internet]. [accessed on 12 march 2023]. Available online: https://www.ajicjournal.org/article/S0196-6553(07)00740-7/fulltext

7- Williams VR, Leis JA, Trbovich P, Agnihotri T, Lee W, Joseph B, et al. Improving healthcare worker adherence to the use of transmission-based precautions through application of human factors design: a prospective multi-centre study. J Hosp Infect. 2019;103(1):101–5.

Reviewer 3 Report

This manuscript has been written very-well. I love it reading. They authors have cent per cent followed all the steps required to conduc a systematic review and have drafted the findings in an excellent way. I just have two minor comments if the authors can adjust may enhance the manuscript further.

First, the authors stated that "
A total of 28 articles (12 of development and 16 of validation) containing 13 156 measurement tools were included in the review", why all the studies were exclusivley conduted either as development or as validation? I am asking this, because I am myself have developed a tool of my own for my doctoral research and it was both development and validation.  So, I think the authors may have missed some of the studies which included both development and validation.

Second, what metrics were used to label the studies as Moderate, low and very low. I read that that the authors have mentioned "In the fourth stage, the scores were summarized for each instrument into an overall rating, and an assessment of the quality of evidence for each instrument (from "high" to "very low") was made on this." This summariaztion is missing, or may be I failed to noticed. I will recommend to include this summarization in tabulated form.

Reviewer 4 Report

REVIEW FOR MANUSCRIPT ENTITLED “EVALUATION OF INSTRUMENTS ASSESSING HEALTH PROFESSIONALS' COMPLIANCE WITH STANDARD PRECAUTIONS: A SYSTEMATIC REVIEW”.

Thank you for providing me with the opportunity to review the manuscript entitled  

“Evaluation of instruments assessing health professionals' compliance with standard precautions: a systematic review”.

Congratulations for the great work done.

After having exhaustively reviewed the following publication, one can appreciate the important scientific work that has been carried out and the great quality of the same.

In this regard, the following scientific paper is suitable for publication.

Author Response

We thank you for the comment and for appreciating our work.